# The Effect of Inspiratory Muscle Training on Health-Related Fitness in College Students

**DOI:** 10.3390/ijerph21081088

**Published:** 2024-08-17

**Authors:** Lili Qin, Siyu Liu, Shuang Hu, Linlin Feng, Huan Wang, Xingzhou Gong, Wei Xuan, Tianfeng Lu

**Affiliations:** Department of Physical Education, Sports and Health Research Center, Tongji University, Shanghai 200092, China; qinlili@tongji.edu.cn (L.Q.); 2231805@tongji.edu.cn (S.L.); 2333369@tongji.edu.cn (S.H.); 10098@tongji.edu.cn (L.F.); 21139@tongji.edu.cn (H.W.); 94774@tongji.edu.cn (X.G.)

**Keywords:** physical fitness, inspiratory muscle training, athletic performance

## Abstract

In an era characterized by rapid economic growth and evolving lifestyles, college students encounter numerous challenges, encompassing academic pressures and professional competition. The respiratory muscle endurance capability is important for college students during prolonged aerobic exercise. Therefore, it is of great significance to explore an effective intervention to enhance the endurance level of college students. This study explores the transformative potential of inspiratory muscle training (IMT) to improve the physical functions of college students. This research comprised a group of 20 participants who underwent IMT integrated into their daily physical education classes or regular training sessions over an 8-week period, with 18 participants forming the control group. The IMT group adhered to the manufacturer’s instructions for utilizing the PowerBreathe device. The findings indicated a significant positive effect on inspiratory muscle strength (*p* < 0.001), showing improvements in pulmonary function, exercise tolerance, cardiac function, and overall athletic performance. These results revealed the substantial benefits of IMT in enhancing physical fitness and promoting health maintenance among college students.

## 1. Introduction

With economic development and lifestyle transitions, the physical fitness of college students has notably declined. The evaluation of physical endurance in college students has yielded disheartening results [1], with male students displaying inferior performance compared to their female counterparts. The low physical fitness levels among college students pose a risk for future disease development, a trend that has worsened globally over the past five decades [2,3]. This underscores a health concern among college students that warrants attention. During high-intensity and prolonged exercise, respiratory muscle strength and performance diminish due to increased respiratory muscle workload and dyspnea [4], ultimately leading to a decline in overall performance. This further illustrates that inadequate endurance capabilities in college students can result in diminished athletic and competitive performance, as well as potentially contributing to overall suboptimal health. Therefore, the endurance levels and physical fitness of university students play a critical role in fostering their holistic development, highlighting the imperative need for targeted training and interventions in the present day.

Accordingly, the hypothesis that training the respiratory muscles, specifically the inspiratory muscles, can enhance exercise endurance and strength has been studied extensively over the past few decades [5,6]. Multiple studies have demonstrated the effectiveness of respiratory muscle training in improving respiratory muscle strength [7,8]. This type of training may enhance resistance to fatigue during full-body exercises [9]. Specifically, inspiratory muscle training (IMT), a component of respiratory muscle training, has emerged as a promising strategy to enhance endurance performance and athletic capabilities across different age groups [9,10]. Inspiratory muscle training is a prevalent component in the training regimens of elite athletes across various sporting disciplines. For instance, in competitive swimming, six weeks of IMT led to enhanced inspiratory strength and performance [11,12]. Elite endurance runners experienced an increase in relative VO_2_max and a decrease in post-training blood lactate levels after eight weeks of IMT [13]. Additionally, soccer players who underwent IMT for eight weeks showed improvements in anaerobic and interval aerobic endurance tests, as well as exercise tolerance [10]. Inspiratory muscle training is a focused exercise technique that can enhance respiratory system function, optimize oxygen utilization efficiency, and improve oxygen delivery by targeting key inspiratory muscles. Examining the impact of IMT on the athletic performance and fitness levels of college students is a valuable direction to address the prevalent issues of low endurance and poor physical conditions in this demographic. 

During upper-extremity exercise, the diaphragm and abdominal muscles co-activate to increase intra-abdominal pressure [14]. Furthermore, during running, the inspiratory muscles of the trunk play a role in maintaining postural stability [15]. Therefore, inspiratory muscle training is crucial for improving individuals’ ability to optimize training benefits and meet the demands of running, ultimately leading to enhanced athletic performance. Current studies have predominantly focused on exploring the effects of inspiratory muscle training on inspiratory muscle strength and exercise capacity in competitive athletes and sports enthusiasts [16,17,18,19,20,21]. In addition to these populations, numerous individuals with medical conditions necessitate enhanced respiratory function and quality of life, such as asthma, and those afflicted with respiratory-related ailments exist [22]. However, the concerning trend of deteriorating physical health in college students is frequently disregarded or inadequately addressed. There is a relative lack of research on the application and effectiveness of inspiratory muscle training in college populations, as well as limited evidence on the effects on endurance qualities and athletic performance in college students.

This study aims to investigate the potential enhancement of respiratory muscle function through specific inspiratory muscle training to improve athletic endurance. The goal is to identify precise and effective interventions that can improve the physical fitness and health of college students. Our research hypothesized that integrating 8 weeks of inspiratory muscle training into daily physical education classes or regular training sessions could improve physical fitness among college students. 

## 2. Materials and Method

### 2.1. Participants

According to the statistical power requirements established in a previous study [23], it is stipulated that a minimum of eight subjects per group is essential. Additionally, the sample size was estimated before the experiment, leading to the recruitment of 42 participants. Prior to randomization, 4 participants withdrew, resulting in a final total of 38 participants. In this study, 38 healthy college students with moderately low scores on a physical fitness assessment were randomly allocated into two groups: an inspiratory muscle training (IMT) group comprising 20 participants and a control group comprising 18 participants. Participants were assigned to the experimental or control group based on their assigned numbers using a random number generator or software. Participants were assigned numbers prior to being allocated to either the experimental or control group through the application of a random number generator or software. The inclusion criteria comprised the following: (1) non-smoking status, (2) aged over 18 years, (3) absence of respiratory or pulmonary pathologies, (4) lack of formal exercise training background and consistent exercise routines, and (5) no usage of supplements or medications that could potentially influence the study outcomes. The experimental procedure received approval from the University Institutional Ethics Committee, and prior to the study’s commencement, participants provided informed consent after receiving comprehensive information about the training procedures and potential associated risks. The procedural flow is delineated in Figure 1.

### 2.2. Experimental Design and Task

Baseline data were collected from all participants before the inspiratory muscle training (IMT) intervention. This study included the assessment of maximal inspiratory pressure, pulmonary function, exercise tolerance, cardiac function, and athletic performance. Assessments were consistently performed at similar times of the day, with the IMT group being assessed first. Participants were instructed to avoid vigorous physical activity, alcoholic beverages, and stimulants (such as coffee, tea, and soft drinks) the day before the tests. They were also advised to stay well-hydrated and have a light meal up to two hours before testing. Prior to the experiment, participants underwent health screenings to confirm their eligibility for the study. Following the intervention, all participants underwent a repeat data collection process.

### 2.3. Measurements

#### 2.3.1. Maximal Inspiratory Pressure (MIP) Assessment

The maximum inspiratory pressure (MIP) assessment in this study utilized the AirOFit PRO™ device (AirOFit, Copenhagen, Denmark), which facilitates the measurement and visualization of breathing patterns through the AirOFit PRO™ Sport mobile app. The device is equipped with parameters for assessing variables like MIP, which not only enhance usability but also offer additional valuable information to the user [24]. Prior to the assessment, each participant completed six preliminary trials to warm up and acclimate to the procedure. Subsequently, six maximum efforts (3 inspiratory and 3 expiratory) with a 45-s rest interval between each trial were performed, and the best result was recorded. Participants were instructed to maintain a lip seal on the mouthpiece, and a nose clip was used to prevent nasal air leakage. Notably, all participants were inexperienced in such testing, mitigating potential learning biases.

#### 2.3.2. Pulmonary Function Assessment

The pulmonary function tests were conducted using a calibrated smart handheld spirometer, following the guidelines set forth by the ATS/ERS Working Group for standardization [25]. These guidelines specify that a minimum of three acceptable maneuvers is necessary for accurate testing. Peak expiratory flow, expiratory lung volume, forced expiratory volume in the first second (FEV1), and forced expiratory flow at one second (FEF1) were evaluated before and after the experiment using a spirometer (Breath Home) compliant with the standards of the European Respiratory Society. The participants remained seated in a standard position during the entire testing procedure. 

#### 2.3.3. Exercise Tolerance and Cardiac Function Assessment

The participants were engaged in a modified incremental load exercise experiment that consisted of multiple stages. Initially, they performed a 30-s 0-load slow-walking acclimatization exercise on a running platform with no incline. This was followed by a 3-min warm-up session at a speed of 6 km/h in a comfortable running position. Subsequently, the participants were fitted with a respiratory mask, and the portable wireless telemetric exercise cardiopulmonary fitness testing system, K5b2, from COSMED in Italy, was connected. The instrument’s measurements are highly reproducible and accurate, allowing for real-time gas metabolism measurement during exercise through breath-by-breath analysis [18,26,27].

The main phase of the experiment started at a speed of 7 km/h, with the speed increasing by 1 km/h per minute until it reached 16 km/h. At this juncture, the speed remained constant until the participants reached exhaustion, at which point the time to exhaustion was recorded. Exhaustion was determined by specific conditions: (1) reaching the individual’s maximum heart rate (HRmax = 208 − 0.7 × age); (2) achieving a respiratory quotient of 1.15 or near it; (3) observing a plateau in oxygen uptake despite increasing exercise intensity; (4) being unable to sustain the prescribed exercise intensity due to extreme fatigue, shortness of breath, dizziness, unsteady gait, and maximal exertion. Exhaustion typically occurs after repeated encouragement and the inability to persevere. Throughout the test, exhaled gases, including oxygen uptake, carbon dioxide excretion, and a respiratory quotient, were collected approximately every three seconds. Simultaneously, heart rate data were recorded using a heart rate receiver.

#### 2.3.4. Athletic Performance Assessment

A 1000-m time trial was conducted both before and after the experiment to assess the impact of the intervention. According to previous research in evaluating athletic performance [28], a suitable metric is needed for assessing the benefits of respiratory muscle training. Participants abstained from intense physical activity for 48 h prior to the 1000-m test and engaged in a 5-min warm-up session before the trial. The trials were conducted on the campus track, with time measurements taken using a counter and oversight provided by two consistent experimental supervisors.

#### 2.3.5. Inspiratory Muscle Training

This study employed PowerBreathe, a respiratory training device with adjustable resistance, produced by PowerBreathe International Ltd. in Southam, Warwickshire, England, UK. The 8-week intervention for the IMT group comprised daily sessions including 2 sets of 30 inhalation repetitions at varying intensities, commencing at 50% MIP and progressing to 80% MIP throughout the intervention. The IMT group also underwent reassessment throughout the measurement of MIP upon completion of the weekly training regimen. The reassessment process incorporated a movement review session, which involved adjustments to training load intensity and instructional support. Participants experiencing breathlessness or discomfort during training were permitted a brief recovery period; nevertheless, they were expected to complete all prescribed repetitions. All training sessions required participants to maintain a correct seated position, use nose clips, keep their lips pressed firmly against the bite, lightly clench their teeth on the pads, and perform rapid, forceful inhalations followed by slow, quiet exhalations. Each participant was directed to diligently record and update a learning log during the entire experiment. During the 8-week study period, the control group did not undergo inspiratory muscle training or receive guided training intervention; instead, they engaged in regular physical education classes and exercise activities.

The experimental design included a standardized instructional training session before the warm-up exercise at the beginning of each exercise session. The focus was on adjusting the warm-up training load to prevent disruption to the exercise routine. The IMT group followed two warm-up protocols: one integrated into the daily physical education class routine, which included a respiratory warm-up before the standard warm-up and another before extracurricular activities. These warm-ups involved inspiratory muscle training techniques with 50% MIP and 30 repetitions per set. Before participating, all participants had to complete a standard instructional training session involving 2 sets of training volume per day. Details of the process are provided in Figure 2.

### 2.4. Statistical Analysis

The sample size was calculated based on the anticipated mean value of maximal inspiratory pressure (MIP) in the control group after the experiment, which was expected to be 99.06 ± 30.74 cm H_2_O. It was projected that the MIP value in the intervention group would increase by 23 cm for H_2_O, with similar variance between the two groups. The two-tailed test α was 0.05, the sample size ratio between the two groups was established at 1:1 (an equal number of participants in each group), and the expected power of the test was set at 90% (1 − β), resulting in a final required sample size of 35 participants. 

The data were analyzed using IBM SPSS Statistics 27 software. Initially, the anthropological information of participants was outlined, and group variances were assessed via independent sample *t*-tests. Normality and chi-square tests were conducted on both groups prior to the experiment. Subsequent hypothesis testing employed independent sample *t*-tests; when assumptions for parametric tests were not met, non-parametric tests, specifically Mann–Whitney U-tests, were utilized to assess the differences between groups. After the experiment, paired-sample *t*-tests were employed to compare data between the experimental and control groups, while Wilcoxon signed-rank tests were used to evaluate within-group differences before and after the intervention. Statistical significance was set at *p* < 0.05. Finally, within-group effect sizes (ESs) for pre- and post-intervention changes were calculated using Cohen’s d. Values equal to or less than 0.20 were classified as “no effect”, values between 0.21 and 0.49, “small effect” values between 0.50 and 0.79, “moderate effect”, and values equal to or greater than 0.80 were classified as “large effect.” [29]. Data are presented as mean ± SD. 

## 3. Results

Table 1 presents the anthropological characteristics of the participants. There were no significant differences between the groups.

### 3.1. Comparative Analysis of Participants’ Pre-Experimental Test Metrics

Before the commencement of the experiment, all participants underwent assessments to evaluate different indicators. Subsequently, an independent sample *t*-test was conducted after consolidating the assessment data. The detailed outcomes are presented in Table 2. 

The analysis of all participant indicators prior to the experiment revealed a lack of variability between the two groups, eliminating the potential for participant characteristics or external factors to influence the study results inaccurately.

### 3.2. Comparative Analysis of Test Metrics before and after the Participants’ Experiment

The IMT group participated in an 8-week didactic training program. Following the experiment, all participants from both groups underwent a secondary assessment, and the outcomes, along with the data analysis, are presented in Table 3.

After 8 weeks of teaching and training, the data in Table 3 were analyzed. The students in the IMT group showed significant improvements in all indicators except for VO_2_max, METS, VE, and BR related to exercise tolerance and cardiac function, which did not exhibit significant changes. MIP values increased significantly from pre- to post-intervention (*p* < 0.001), with effect size (ES) values appearing larger in the IMT group and smaller in the control group. However, the effect size (ES) values for FVC, FEV1, and FEV1/FVC were moderate in the IMT group and small in the control group. The IMT group demonstrated a significant increase in pulmonary function, while no significant changes were noted in the control group (Figure 3). These results suggest a notable enhancement in the exercise capacity of the students. Conversely, the control group, who received standard teaching, did not show significant differences for most indicators, with the exceptions being VO_2_max, METS, RQ, and VO_2_/HR (*p* < 0.05) pertaining to exercise tolerance and cardiac function. ES values were similar in both groups but larger in the IMT group. Due to factors such as variations in body types that could have influenced the VO_2_max results, there were discrepancies between the observed VO_2_max results in the IMT group and the control group when compared to the initial predictions. Interestingly, a notable change in VO_2_max was noted in the control group, while the IMT group did not exhibit significant alterations as anticipated. Participants in the IMT training group experienced additional benefits, including a significant improvement in *t*-test values (Figure 4). In the selected 1000 m tests, the IMT group demonstrated a significant change (*p* < 0.001; ES = 0.633), while no significant differences were found in the control group from pre-intervention to post-intervention (*p* > 0.05; ES = 0.111).

## 4. Discussion

The purpose of this study was to evaluate the effects of an eight-week inspiratory muscle training (IMT) program added to a regular physical education curriculum on maximal inspiratory pressure, lung function, and athletic performance in college students. The results indicate that the systematic IMT program significantly enhanced maximal inspiratory pressure and improved lung function, which directly correlated with increased endurance and athletic performance in the trained group. Conversely, the control group, which did not undergo the IMT, exhibited no significant improvements, with only minor increases in a few parameters. This study not only improved students’ fitness and health but also provided valuable insights for the development of physical education curricula and teaching methodologies. 

### 4.1. Effects of Inspiratory Muscle Training in Maximal Inspiratory Pressure and Pulmonary Function Assessment

A significant increase in maximal inspiratory pressure was observed after 8 weeks of inspiratory muscle training (IMT), with values rising from 100.50 ± 27.30 cm H_2_O to 150.65 ± 28.51 cmH_2_O. Aznar-Lain et al. reported a significant improvement in maximal inspiratory pressure (MIP) values following IMT training [30]. Additionally, a study on soccer players demonstrated the benefits of IMT in enhancing MIP among professional athletes [20]. In our study, we observed a substantial enhancement in MIP among students, potentially due to the initial weakness of their respiratory muscles prior to IMT application. Volianitis et al. implemented an intriguing experimental design wherein rowers in the IMT group and placebo group underwent different training regimens. The results showed a greater increase in MIP among athletes in the IMT group compared to the placebo group, although both groups exhibited significant improvements in exercise endurance [31]. These findings suggest that respiratory muscle training can significantly enhance a student’s physical education experience, with an 8-week program focusing on respiratory muscle training leading to improved endurance and athleticism. This novel approach presents new opportunities for physical education programs. Students in the respiratory muscle training group displayed notable enhancements in lung function indicators, with peak expiratory flow (PEF) serving as a key metric. Utilizing respiratory muscle training as a resistance program can effectively improve PEF [18,32] and further validate these results. In addition, only the IMT group showed a significant improvement in FEV1, FVC, and the Tiffeneau index (FEV1/FVC) (*p* < 0.05). Therefore, the current study adhered to the spirometer protocol guidelines and norms during spirometry procedures. Emphasis was placed on forceful exhalation, in line with common practice among spirometry technicians, and participants were encouraged to take a deep inhalation before performing the exhalation maneuver, resulting in improved outcomes.

According to the literature, the duration of training significantly impacts the rate of maximal inspiratory pressure (MIP) growth. Physiologically, in order to fully realize the effects of training, neural adaptations must be translated into functional changes in musculature, necessitating an 8-week training cycle [33]. Consistent with prior research, our study validates these findings. Hoffman et al. explored the connection between training frequency and muscle strength, observing that strength training conducted up to 5 days per week resulted in a notable increase in muscle strength, indicating a strong correlation between frequency and efficacy. In our study, we implemented a training regimen of 6 days per week to enhance the students’ capabilities. With the expanding scholarly knowledge on respiratory muscle training, there is a growing consensus that longer durations and higher intensities are essential for the general population to achieve substantial training outcomes. To tackle this issue, we extended the experimental duration and progressively elevated the training intensity.

### 4.2. Effects of Inspiratory Muscle Training on Exercise Tolerance and Cardiac Function Assessment

This study illustrated that an eight-week didactic training program led to enhancements in exercise endurance and cardiac function. Retrospective analysis indicated that respiratory muscle improvement predominantly impacts endurance and cardiorespiratory fitness [34]. Mickleborough et al. discovered that respiratory muscle training modifies respiratory mechanics and enhances oxygen consumption capacity, ventilation, heart rate, and perceptual responses during blood lactate concentration and sustained workload exercises [23]. In our study, changes in heart rate before and after the test were the only observable differences among the various factors analyzed. The heart rate measured after test completion did not exhibit any significant variances between the groups despite a notable enhancement in running duration observed in the IMT group. These results suggest a potential enhancement in running the economy as a result of inspiratory muscle training. Additionally, it enhances endurance exercise performance in runners. Among these benefits, maximal oxygen uptake is a crucial determinant of aerobic endurance performance and aids in reducing the body’s burden during prolonged exercise and work. In our study, the IMT group did not exhibit significant improvements post-training, whereas notable enhancements were observed in the control group. Nevertheless, there is no conclusive evidence supporting the direct enhancement of maximal oxygen uptake through inspiratory muscle training. It is important to consider that improvements in VO_2_max are influenced by various factors, including genetic predisposition and individual variances. Higher maximal oxygen uptake does not necessarily translate to superior athletic performance, as success in endurance sports is influenced by a myriad of factors such as body composition, neural efficiency, muscle strength, technical proficiency, economy of motion, and years of athletic training experience. Morgan et al. proposed that respiratory muscle training enhanced ventilation and endurance in moderately trained male cyclists without affecting their maximal oxygen uptake or endurance cycling performance. Kilding et al. discovered that club-level-trained swimmers experienced fewer improvements in swimming performance for races shorter than 400 m after completing a 6-week inspiratory muscle training (IMT) program [11]. Sperlich et al. observed no significant alterations in maximal oxygen uptake or endurance performance following high-intensity respiratory muscle training (RMT) among the special forces and military personnel [35].

Differences in these findings may arise from various factors, such as the training methods, protocols, equipment used, load configurations, and limited sample sizes. While inspiratory muscle training is known to offer significant benefits in enhancing endurance performance, the specific mechanisms behind these improvements require further exploration. Nicks et al. observed that RMT enhanced intermittent exercise performance in soccer players [36]; however, additional research is necessary to investigate the mechanisms through which it enhances performance. Improving running economy is essential for achieving superior endurance running performance. Subsequent studies should track participants in the inspiratory muscle training group to understand the impact of inspiratory muscle training on athletic performance, particularly for endurance capacity. Additionally, testing and guided training sessions could be repeated after a guided training phase followed by an unguided phase. This study introduces new concepts that can benefit a population of healthy college students with low fitness levels, suggesting a combination of physical education classes and IMT training to enhance overall endurance levels.

### 4.3. Effects of Inspiratory Muscle Training in Athletic Performance Assessments

This study employed a visual measure of athletic performance, specifically focusing on the 1000-m running time of participants. The results indicate a significant decrease in time for the respiratory muscle training group, from 265.15 ± 19.60 s to 252.55 ± 20.21 s. This finding aligns with a similar study involving college students who underwent four weeks of respiratory muscle training, showing enhancements in both respiratory muscle strength and 800-m exercise performance. This consistency underscores the agreement between our study’s outcomes and those of prior research. Tong et al. suggested that incorporating a blend of chronic and acute inspiratory muscle training into a high-intensity interval running regimen can effectively enhance tolerance to high-intensity intermittent running [37]. Another study also observed that inspiratory muscle training, along with warm-up exercises, contributed to improving tolerance to strenuous intermittent exercise [38]. Therefore, inspiratory muscles offer significant advantages in boosting overall exercise performance.

The application of respiratory muscle training in the college population involves implementing various strategies to enhance athletic performance and health. These strategies include developing personalized training programs, integrating different forms of training, emphasizing correct breathing techniques, advocating for professional respiratory muscle training equipment, conducting regular assessments and monitoring, and promoting consistent training. As an educator, it is essential to update teaching methods to support the healthy development of college students. 

The strength of this study lies in the selection of the participants and the evaluation of the IMT intervention effect. By specifically targeting this population for inspiratory muscle training, this study aims to elucidate the role of these muscles in enhancing athletic performance. This research not only contributes to the advancement of scientific knowledge in the field but also holds promise for developing tailored training programs to enhance fitness among college students. However, several limitations need to be addressed. Firstly, extending the duration of the study and expanding the sample size is crucial. Future research endeavors should also encompass diverse demographics, including various races, ages, genders, and cultures, to ensure the generalizability of the findings. Secondly, the continuous exploration of the effects of inspiratory muscle training on overall health and athletic performance is essential for a comprehensive understanding of its benefits. Lastly, integrating inspiratory muscle training with other training modalities is recommended to investigate potential synergistic effects and optimize training outcomes. Some limitations of our study must be taken into account when interpreting the experiment’s conclusions. The study period should be prolonged, and the sample size should be increased. Future research should explore the impact of inspiratory muscle training on college students’ athletic performance, distinguishing between speed and endurance training and examining whether it enhances endurance performance by reducing dyspnea.

## 5. Conclusions

Eight weeks of inspiratory muscle training (IMT) have been shown to enhance students’ respiratory muscle strength, lung function, exercise tolerance, and cardiac function. Integrating IMT into physical education programs on campus can be beneficial in stimulating student engagement and enhancing their physical fitness levels. Subsequent research should include a follow-up assessment of participants in the inspiratory muscle training group to understand the impact of this training on athletic performance, particularly for endurance capacity. Furthermore, testing and guided training sessions should be repeated after a guided training phase followed by an unguided phase. Emphasis should also be placed on evaluating the potential applicability of supplementary training to a wider student population to maximize this study’s benefits.

## Figures and Tables

**Figure 1 ijerph-21-01088-f001:**
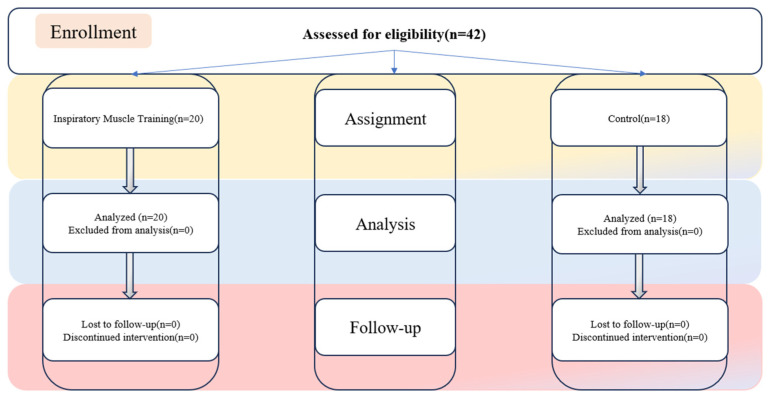
Participant selection and randomization process.

**Figure 2 ijerph-21-01088-f002:**
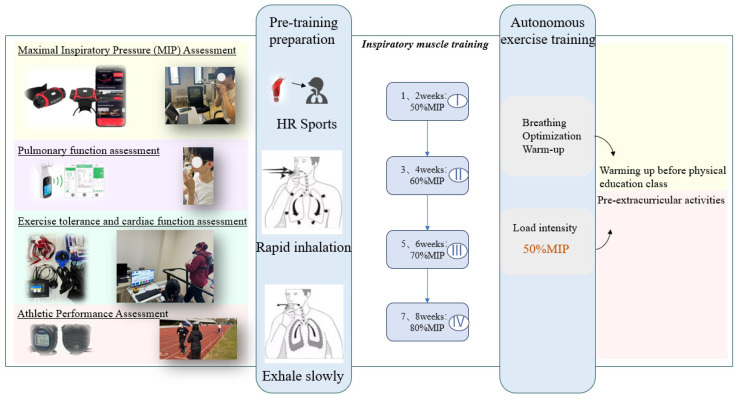
Testing and training process.

**Figure 3 ijerph-21-01088-f003:**
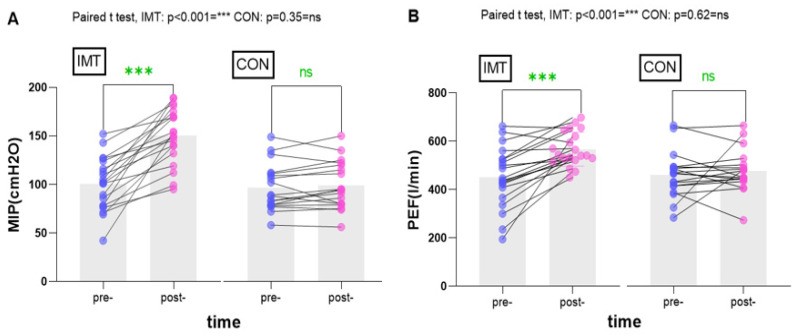
Effects of inspiratory muscle training (IMT) on (**A**) maximal inspiratory pressure(MIP) and (**B**) peak expiratory flow(PEF).

**Figure 4 ijerph-21-01088-f004:**
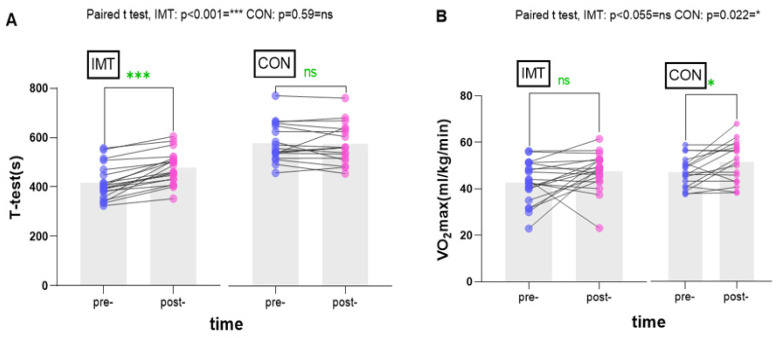
Effects of inspiratory muscle training (IMT) on (**A**) exercise tolerance and (**B**) cardiac function assessment.

**Table 1 ijerph-21-01088-t001:** Detailed information of the participants (n = 38).

Anthropological Information	IMT Group (n = 20) Mean ± SD	Control Group (n = 18) Mean ± SD	*p*
Age, years	18.55 ± 0.89	18.67 ± 1.08	0.74
Height, cm	176.9 ± 6.48	176.06 ± 5.6	0.67
Mass, kg	68.65 ± 10.63	68.39 ± 12.5	0.95
BMI	21.98 ± 3.59	22.00 ± 4.40	0.93

**Table 2 ijerph-21-01088-t002:** Comparison of test index data between experimental and control groups.

Variables	IMT Group (n = 20)	Control Group (n = 18)	t/Z	*p*
	Pre-	Pre-		
**MIP**				
MIP (cmH_2_O)	100.50 ± 27.30	96.72 ± 24.33	0.448	0.66
**pulmonary function**				
PEF (L/min)	451.10 ± 124.13	460.39 ± 96.04	−0.256	0.80
PEF (%)	70.65 ± 19.50	72.11 ± 14.97	−0.073	0.942
FVC (L)	3.30 ± 0.906	2.90 ± 0.94	1.307	0.199
FEV1 (L)	3.28 ± 0.90	2.82 ± 0.85	1.597	0.119
FEV1 (%)	63.65 ± 15.75	58.17 ± 17.53	1.016	0.316
FEV1/FVC (%)	95.30 ± 8.85	97.94 ± 3.49	−0.550	0.583
**Exercise tolerance**				
VO_2_max (mL/min/kg)	42.72 ± 9.03	47.08 ± 6.54	−1.685	0.10
METS	12.22 ± 2.59	13.44 ± 1.85	−1.667	0.104
RQ	1.12 ± 0.10	1.05 ± 0.1	−1.934	0.053
VE (L/min)	77.62 ± 26.74	73.27 ± 18.44	0.576	0.57
BR (%)	53.41 ± 15.74	55.65 ± 11.03	−0.501	0.62
HRmax (bpm)	190.25 ± 16.29	189.89 ± 8.67	−0.834	0.404
VO_2_/HR (mL/beat)	15.10 ± 2.22	16.92 ± 3.45	−1.579	0.114
*t*-test (s)	417.00 ± 70.19	433.67 ± 58.52	−1.023	0.306
**Athletic Performance**				
1000 m test (s)	265.15 ± 19.60	259.28 ± 38.60	−0.951	0.342

Abbreviations: MIP = maximum inspiratory pressure; PEF = peak expiratory flow; FVC = forced vital capacity; FEV1 = forced expiratory volume in the first second; VO_2_max = maximum oxygen uptake; METS = metabolic equivalent of task; RQ = respiratory quotient; VE = minute ventilation at corresponded time; BR = breathing reserve; and HR = heart rate.

**Table 3 ijerph-21-01088-t003:** Comparison of participant test indicator data before and after the experiment.

	IMT Group		t/Z	ES	*p*	Control Group		t/Z	ES	*p*
	Pre-	Post-				Pre-	Post-			
MIP (cmH_2_O)	100.50 ± 27.30	150.65 ± 28.51	−3.921	1.797	<0.001	96.72 ± 24.33	99.06 ± 24.74	−0.953	0.095	0.354
PEF (L/min)	451.10 ± 124.13	567.25 ± 70.54	−3.734	1.151	<0.001	460.39 ± 96.04	477.17 ± 88.97	−0.501	0.181	0.616
PEF (%)	70.65 ± 19.50	90.60 ± 11.78	−3.772	1.238	<0.001	72.11 ± 14.97	76.11 ± 13.99	−0.640	0.276	0.522
FVC (L)	3.30 ± 0.906	2.75 ± 0.91	2.582	0.606	0.018	2.90 ± 0.94	2.80 ± 0.98	0.536	0.104	0.599
FEV1 (L)	3.28 ± 0.90	2.75 ± 0.91	2.506	0.586	0.021	2.82 ± 0.85	2.80 ± 0.98	0.136	0.22	0.893
FEV1 (%)	63.65 ± 15.75	58.25 ± 19.64	1.264	0.303	0.222	58.17 ± 17.53	57.50 ± 18.43	0.188	0.037	0.853
FEV1/FVC (%)	95.30 ± 8.85	100 ± 0.00	−2.371	0.751	0.018	97.94 ± 3.49	98.22 ± 5.86	−0.405	0.058	0.686
VO_2_max (mL/min/kg)	42.73 ± 9.03	47.58 ± 8.08	−2.041	0.566	0.055	47.08 ± 6.54	51.58 ± 8.76	−2.291	0.582	0.022
METS	12.22 ± 2.59	13.60 ± 2.31	−2.030	0.562	0.057	13.44 ± 1.85	14.74 ± 2.66	−2.546	0.567	0.021
RQ	1.12 ± 0.10	1.01 ± 0.07	4.786	1.274	<0.001	1.05 ± 0.1	0.94 ± 0.08	−3.468	1.215	<0.001
VE (L/min)	77.62 ± 26.74	85.17 ± 20.84	−1.016	0.315	0.323	73.27 ± 18.44	71.63 ± 19.89	0.534	0.086	0.60
BR (%)	53.41 ± 15.74	49.09 ± 12.44	0.993	0.305	0.333	55.65 ± 11.03	59.19 ± 13.32	−1.034	0.289	0.301
HRmax (bpm)	190.25 ± 16.29	198.00 ± 10.04	−2.266	0.573	0.023	189.89 ± 8.67	191.44 ± 7.30	−1.514	0.193	0.13
VO_2_/HR (mL/beat)	15.10 ± 2.22	16.47 ± 3.49	−1.868	0.468	0.062	16.92 ± 3.45	18.28 ± 3.97	−2.587	0.366	0.019
*t*-test (s)	417.00 ± 70.19	478.05 ± 65.11	−3.920	0.902	<0.001	433.67 ± 58.52	431.89 ± 59.98	−0.545	0.03	0.586
1000 m test (s)	265.15 ± 19.60	252.55 ± 20.21	5.042	0.633	<0.001	259.28 ± 38.60	263.22 ± 32.35	−1.527	0.111	0.127

Abbreviations: MIP = maximum inspiratory pressure; PEF = peak expiratory flow; FVC = forced vital capacity; FEV1 = forced expiratory volume in the first second; VO_2_max = maximum oxygen uptake; METS = metabolic equivalent of task; RQ = respiratory quotient; VE = minute ventilation at corresponded time; BR = breathing reserve; and HR = heart rate.

## Data Availability

The data presented in this study are available on request from the corresponding author.

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
