# Peer review of "The Effect of Inspiratory Muscle Training on Health-Related Fitness in College Students"

_ijerph, 2024, doi:10.3390/ijerph21081088_

Round 1
Reviewer 1 Report
Comments and Suggestions for Authors
Dear Authors, the document entitled “The Effect of Inspiratory Muscle Training on Health-Related Fitness in College Students: Diversification of extracurricular physical activity” is very interesting, original, and easy to read. Some suggestions are made to improve the overall quality of the document.
I would suggest you remove ": Diversification of extracurricular physical activity" from the title. Moreover, at the end of the introduction chapter, I would suggest the removal of the following section “The cultivation of college students' awareness of physical fitness and the promotion of an active and healthy lifestyle among them can contribute to fostering a vibrant and health-conscious cultural atmosphere throughout society.”, it might be considered speculative and does not add to the aim and objectives of the present study.
In your materials and methods section, if you could please add the sample size calculation, It would improve the reliability of the information provided by this investigation. Furthermore, please explain the randomization method and whether blinding of the participants/investigators was guaranteed. I would suggest that you describe more the reliability, validity and minimal detectable change of all the included instruments. Moreover, please extensively explain the protocol for the control group.
In the results section, I would suggest that you add information regarding the statistical test being applied to each table. I can see from Table 1 that perhaps obese participants were included in the study, it might be an important bias since obesity impairs respiratory performance, please consider this aspect. Please explain more about the effects described in Figures 3 and 4, is it effect size? Would you please inform the effect sizes for all the significant measures?! Table 4 please include a legend to explain the meaning of “df”.
At the end of your discussion section please remove the following: “The authors should analyze the study results and their interpretation about previous research and initial hypotheses. The implications of the findings should be discussed in a broad context, and potential future research directions should be highlighted. ”
Please include the limitations of your work at the end of the discussion section, remove the indications for future studies from the discussion section, and send it to the end of the conclusions section.
Author Response
- I would suggest you remove ": Diversification of extracurricular physical activity" from the title. Moreover, at the end of the introduction chapter, I would suggest the removal of the following section “The cultivation of college students' awareness of physical fitness and the promotion of an active and healthy lifestyle among them can contribute to fostering a vibrant and health-conscious cultural atmosphere throughout society.”, it might be considered speculative and does not add to the aim and objectives of the present study.
Response: Thank you for reviewer’s valuable and thoughtful comments, we have modified the corresponding part in the manuscript. Firstly, the title has been modified as “The Effect of Inspiratory Muscle Training on Health-Related Fitness in College Students”. Secondly, at the end of the introduction chapter, we have removed the section according to the reviewer’s suggestions.
- In your materials and methods section, if you could please add the sample size calculation, It would improve the reliability of the information provided by this investigation. Furthermore, please explain the randomization method and whether blinding of the participants/investigators was guaranteed. I would suggest that you describe more the reliability, validity and minimal detectable change of all the included instruments. Moreover, please extensively explain the protocol for the control group.
Response: Regarding the reviewer’s question about the calculation of the supplementary sample size, we have added some support from the literature in the participant section of the article, regarding previous studies, the number of subjects required for each group is at least 8. Therefore, the number of people set for this study is more reasonable, and it is perfectly fine to add some more sample size supplements in the future. Some explanations were added to related paragraph. Please check the updated Line 86-90.
[1] Mickleborough, T.D.; Nichols, T.; Lindley, M.R.; Chatham, K.; Ionescu, A.A. Inspiratory flow resistive loading improves respiratory muscle function and endurance capacity in recreational runners. Scand J Med Sci Sports 2010, 20, 458-468, doi:10.1111/j.1600-0838.2009.00951.x.
Response: Thanks for your thoughtful comments, we have added the explanations in the participant section of the article that the randomization methodology of this study which was strictly adhered to in the actual study. Some explanations were added to related paragraph. Please check the updated Line 90-94.
Response: The selection of measurement instruments for this study was outlined in a previous manuscript, and we have added some references to ensure the reliability and validity for the instruments included. Some explanations were added to related paragraph. Please check the updated Line 118-120、128-130、143-145、160-162.
[1] Stavrou, V.T.; Tourlakopoulos, K.N.; Daniil, Z.; Gourgoulianis, K.I. Respiratory Muscle Strength: New Technology for Easy Assessment. Cureus 2021, 13, e14803, doi:10.7759/cureus.14803.
[2] Miller, M.R.; Hankinson, J.; Brusasco, V.; Burgos, F.; Casaburi, R.; Coates, A.; Crapo, R.; Enright, P.; van der Grinten, C.P.; Gustafsson, P.; et al. Standardisation of spirometry. Eur Respir J 2005, 26, 319-338, doi:10.1183/09031936.05.00034805.
[3] El-Deen, H.A.B.; Alanazi, F.S.; Ahmed, K.T. Effects of inspiratory muscle training on pulmonary functions and muscle strength in sedentary hemodialysis patients. Journal of Physical Therapy Science 2018, 30, 424-427, doi:10.1589/jpts.30.424.
[4] Guidetti, L.; Meucci, M.; Bolletta, F.; Emerenziani, G.P.; Gallotta, M.C.; Baldari, C. Validity, reliability and minimum detectable change of COSMED K5 portable gas exchange system in breath-by-breath mode. PLoS One 2018, 13, e0209925, doi:10.1371/journal.pone.0209925.
[5] Perez-Suarez, I.; Martin-Rincon, M.; Gonzalez-Henriquez, J.J.; Fezzardi, C.; Perez-Regalado, S.; Galvan-Alvarez, V.; Juan-Habib, J.W.; Morales-Alamo, D.; Calbet, J.A.L. Accuracy and Precision of the COSMED K5 Portable Analyser. Front Physiol 2018, 9, 1764, doi:10.3389/fphys.2018.01764.
[6] Chang, Y.-C.; Chang, H.-Y.; Ho, C.-C.; Lee, P.-F.; Chou, Y.-C.; Tsai, M.-W.; Chou, L.-W. Effects of 4-Week Inspiratory Muscle Training on Sport Performance in College 800-Meter Track Runners. Medicina 2021, 57, doi:10.3390/medicina57010072.
Response: We have added some more details in the part of Inspiratory Muscle Training Intervention,which describes the control group who did not perform inspiratory muscle training during the 8-week study period. Some explanations were added to related paragraph. Please check the updated Line 180-183.
- In the results section, I would suggest that you add information regarding the statistical test being applied to each table. I can see from Table 1 that perhaps obese participants were included in the study, it might be an important bias since obesity impairs respiratory performance, please consider this aspect. Please explain more about the effects described in Figures 3 and 4, is it effect size? Would you please inform the effect sizes for all the significant measures?Table 4 please include a legend to explain the meaning of “df”.
Response: The information about the statistical tests shown in Tables 2 and 3 only showed the effect values, with the aim of being able to observe the training effects more visually. However, the article utilizes two main methods of data analysis to avoid a monolithic approach, and it focuses on the numerical value of the effect force, which facilitates a more intuitive observation of the training effect.
Response: In this study, firstly, the obese participants were excluded according to the criteria of BMI(<28.0kg/m2), secondly, the students who scored moderately low on the physical fitness test were selected, and finally, the method of random sampling set up by the study was executed to determine the experimental and control groups. Please check the updated. Some explanations were added to related paragraph. Please check the updated Line 86-91.
Response: According to the reviewer’s suggestion, we have added that content to clarify what the picture represents. Please check the updated. Some explanations were added to related paragraph. Please check the updated Line 244-250.
Response: Based on the modifications you have suggested, we would like to explain to you that Table 4 is the multivariate ANOVA table adopted for the study. The meaning of df is degrees of freedom, which can be expressed as intergroup degrees of freedom, and intragroup degrees of freedom, respectively.
- At the end of your discussion section please remove the following: “The authors should analyze the study results and their interpretation about previous research and initial hypotheses. The implications of the findings should be discussed in a broad context, and potential future research directions should be highlighted.”
Response: Thank you for your suggestions, we have removed the corresponding part in the discussion section.
- Please include the limitations of your work at the end of the discussion section, remove the indications for future studies from the discussion section, and send it to the end of the conclusions section.
Response: Thank you for your valuable comments. We have revised the discussion section for the limitations and the future studies. Please check the updated. Some explanations were added to related paragraph. Please check the updated Line 400-408、416-419.

Reviewer 2 Report
Comments and Suggestions for Authors
Dear editor, thank you for choosing me as a reviewer. The present study examines the effect of respiratory muscle training on fitness. A practical and very effective research in sports science. In the introduction, the working method should be clearly shown. There is some repetition in the introduction and abstract, it is better to change it. The necessity and importance of conducting research has not been well demonstrated. What was the sampling method? The reference of functional tests is not mentioned in the text The reason for improvement or lack of improvement should be discussed comprehensively
Author Response
In the introduction, the working method should be clearly shown. There is some repetition in the introduction and abstract, it is better to change it. The necessity and importance of conducting research has not been well demonstrated. The reference of functional tests is not mentioned in the text. The reason for improvement or lack of improvement should be discussed comprehensively.
Response: Thank you for the suggestion that the introductory section needs to clearly express the methodology of the work, and we have intentionally added a paragraph to clarify the research. Some explanations were added to related paragraph. Please check the updated Line 78-82.
Response: Regarding as the abstract, we have adjusted that section to present the Abstract section in a concise and precise way, and present the strengths of the study at the end of the Abstract. Some explanations were added to related paragraph. Please check the updated Line 8-22.
Response: Thank you for your suggestions, we have added that section separately in the introduction. Some explanations were added to related paragraph. Please check the updated Line 71-82.
What was the sampling method? The reference of functional tests is not mentioned in the text.
Response: Thanks for your thoughtful comments, we have added the explanations in the participant section of the article that the randomization methodology of this study which was strictly adhered to in the actual study. Some explanations were added to related paragraph. Please check the updated Line 86-91.
Response: According to the reviewer’s suggestions, the selection of measurement instruments for this study was outlined in a previous manuscript, and we have added some references in the part of the functional tests. Some explanations were added to related paragraph. Please check the updated Line 118-120、128-130、143-145、160-162.
[1] Stavrou, V.T.; Tourlakopoulos, K.N.; Daniil, Z.; Gourgoulianis, K.I. Respiratory Muscle Strength: New Technology for Easy Assessment. Cureus 2021, 13, e14803, doi:10.7759/cureus.14803.
[2] Miller, M.R.; Hankinson, J.; Brusasco, V.; Burgos, F.; Casaburi, R.; Coates, A.; Crapo, R.; Enright, P.; van der Grinten, C.P.; Gustafsson, P.; et al. Standardisation of spirometry. Eur Respir J 2005, 26, 319-338, doi:10.1183/09031936.05.00034805.
[3] El-Deen, H.A.B.; Alanazi, F.S.; Ahmed, K.T. Effects of inspiratory muscle training on pulmonary functions and muscle strength in sedentary hemodialysis patients. Journal of Physical Therapy Science 2018, 30, 424-427, doi:10.1589/jpts.30.424.
[4] Guidetti, L.; Meucci, M.; Bolletta, F.; Emerenziani, G.P.; Gallotta, M.C.; Baldari, C. Validity, reliability and minimum detectable change of COSMED K5 portable gas exchange system in breath-by-breath mode. PLoS One 2018, 13, e0209925, doi:10.1371/journal.pone.0209925.
[5] Perez-Suarez, I.; Martin-Rincon, M.; Gonzalez-Henriquez, J.J.; Fezzardi, C.; Perez-Regalado, S.; Galvan-Alvarez, V.; Juan-Habib, J.W.; Morales-Alamo, D.; Calbet, J.A.L. Accuracy and Precision of the COSMED K5 Portable Analyser. Front Physiol 2018, 9, 1764, doi:10.3389/fphys.2018.01764.
[6] Chang, Y.-C.; Chang, H.-Y.; Ho, C.-C.; Lee, P.-F.; Chou, Y.-C.; Tsai, M.-W.; Chou, L.-W. Effects of 4-Week Inspiratory Muscle Training on Sport Performance in College 800-Meter Track Runners. Medicina 2021, 57, doi:10.3390/medicina57010072.
The reason for improvement or lack of improvement should be discussed comprehensively.
Response: Thanks for your thoughtful comments, we have refined the discussion section and added the limitations of the article's research and filled in the closing section with insights for future research. Some explanations were added to related paragraph. Please check the updated Line 309-314、335-340、395-408.

Round 2
Reviewer 1 Report
Comments and Suggestions for Authors
Dear authors, thank you for providing a revised version of your manuscript.
I would recommend that you remove the following “It is proved that IMT…”, from the end of the introduction chapter. The introduction chapter should not have results/conclusions.
Furthermore, for sample size estimation, researchers need to (1) provide information regarding the statistical analysis to be applied, (2) determine acceptable precision levels, (3) decide on study power, (4) specify the confidence level, and (5) determine the magnitude of practical significance differences (effect size), so the information you have provided seem insufficient.
Please provide more information on the procedure for the control group. What do you mean by “a standard exercise regimen”? Please describe.
Please provide information regarding effect size.
The legend of Table 4 seems incomplete. Tables 4 and 5, both multivariate ANOVA and one-way ANOVA are not reported in the standard form. For example, of relevant information to be reported: A one-way multivariate analysis of variance was run to determine the effect of pupils' previous schooling on academic performance. Two measures of academic performance were assessed: English and math's end-of-year exam scores. Pupils arrived from three previous schools: School A, School B and School C. Preliminary assumption checking revealed that data was normally distributed, as assessed by Shapiro-Wilk test (p > .05); there were no univariate or multivariate outliers, as assessed by boxplot and Mahalanobis distance (p > .001), respectively; there were linear relationships, as assessed by scatterplot; no multicollinearity (r = .393, p = .002); and there was homogeneity of variance-covariance matrices, as assessed by Box's M test (p = .003). Pupils in Schools A, B and C scored higher in their English exam (M = 75.6, SD = 8.2; M = 63.6, SD = 6.6 and M = 59.8, SD = 4.6, respectively) than their maths exam (M = 43.9, SD = 8.5; M = 40.8, SD = 8.2 and M = 30.8, SD = 7.7, respectively). The differences between the schools on the combined dependent variables was statistically significant, F(4, 112) = 17.675, p < .001; Wilks' Λ = .376; partial η2 = .387. Follow-up univariate ANOVAs showed that both English scores (F(2, 57) = 30.875, p < .001; partial η2 = .520) and maths scores (F(2, 57) = 14.295, p < .001; partial η2 = .334.) were statistically significantly different between the pupils from different previous schools, using a Bonferroni adjusted α level of .025. Tukey post-hoc tests showed that for English scores, pupils from School A had statistically significantly higher mean scores than pupils from either School B (p < .001) or School C (p < .001), but not between School B and School C (p = .169). For maths scores, Tukey post-hoc tests showed that School C had statistically significantly lower mean scores than pupils from either School A (p < .001) or School B (p = .001).
After “however, several limitations need to be addressed”. Please list all the limitations of your study.
Kind regards.
Author Response
I would recommend that you remove the following “It is proved that IMT…”, from the end of the introduction chapter. The introduction chapter should not have results/conclusions.
Response: We appreciate the reviewer's valuable and thoughtful comments; consequently, we have revised the paragraph accordingly.
Furthermore, for sample size estimation, researchers need to (1) provide information regarding the statistical analysis to be applied, (2) determine acceptable precision levels, (3) decide on study power, (4) specify the confidence level, and (5) determine the magnitude of practical significance differences (effect size), so the information you have provided seem insufficient.
Response: Thanks for the reviewer’s comments. Before the experiment, we evaluated the required sample size using a completely randomized method to divide the subjects into two groups: the inspiratory muscle training (IMT) group and the control group. The primary outcome measure was the maximum inspiratory pressure (MIP) value. Based on prior experience from a pre-experiment and previous studies, we anticipated that the mean MIP value for the control group after the experiment would be 99.06 ± 30.74 cmH₂O. We expected the MIP value for the IMT group to increase by 23 cmH₂O following the intervention, with similar variances between the two groups. A two-tailed test a was 0.05, and the sample size ratio for the two groups was set at 1:1 (an equal number of participants in each group), The desired power of the test was 90% (1-β), calculated according to the appropriate formula. According to formula N= (Za+Zβ)2*2s2/d , N= (1.96+1.28)2*2*30*30/23*23=35, it can be concluded that the required sample size is 35. After estimating the sample size prior to the experiment and accounting for potential participant dropouts and data loss, we added 7 additional participants, bringing the total to 42. Additionally, the sample size was estimated before the experiment, leading to the recruitment of 42 participants. Prior to randomization, 4 participants withdrew, resulting in a final total of 38 participants. Some explanations were added to related paragraph. Please check the updated Line 84-87、196-202.
Please provide more information on the procedure for the control group. What do you mean by “a standard exercise regimen”? Please describe.
Response: Thanks for the reviewer’s comments. The control group participated in regular physical education classes (courses organized on campus) and exercise activities without the inclusion of inspiratory muscle training. Consequently, we have made modifications to clarify the program undertaken by the control group. Some explanations were added to related paragraph. Please check the updated Line 180-183.
Please provide information regarding effect size.
Response: Thanks for the reviewer’s comments, we have included additional information regarding effect sizes.
The legend of Table 4 seems incomplete. Tables 4 and 5, both multivariate ANOVA and one-way ANOVA are not reported in the standard form. For example, of relevant information to be reported: A one-way multivariate analysis of variance was run to determine the effect of pupils' previous schooling on academic performance. Two measures of academic performance were assessed: English and math's end-of-year exam scores. Pupils arrived from three previous schools: School A, School B and School C. Preliminary assumption checking revealed that data was normally distributed, as assessed by Shapiro-Wilk test (p > .05); there were no univariate or multivariate outliers, as assessed by boxplot and Mahalanobis distance (p > .001), respectively; there were linear relationships, as assessed by scatterplot; no multicollinearity (r = .393, p = .002); and there was homogeneity of variance-covariance matrices, as assessed by Box's M test (p = .003). Pupils in Schools A, B and C scored higher in their English exam (M = 75.6, SD = 8.2; M = 63.6, SD = 6.6 and M = 59.8, SD = 4.6, respectively) than their maths exam (M = 43.9, SD = 8.5; M = 40.8, SD = 8.2 and M = 30.8, SD = 7.7, respectively). The differences between the schools on the combined dependent variables was statistically significant, F(4, 112) = 17.675, p < .001; Wilks' Λ = .376; partial η2 = .387. Follow-up univariate ANOVAs showed that both English scores (F (2, 57) = 30.875, p < .001; partial η2 = .520) and maths scores (F (2, 57) = 14.295, p < .001; partial η2 = .334.) were statistically significantly different between the pupils from different previous schools, using a Bonferroni adjusted α level of .025. Tukey post-hoc tests showed that for English scores, pupils from School A had statistically significantly higher mean scores than pupils from either School B (p < .001) or School C (p < .001), but not between School B and School C (p = .169). For maths scores, Tukey post-hoc tests showed that School C had statistically significantly lower mean scores than pupils from either School A (p < .001) or School B (p = .001).
Response: Thanks for the reviewer’s comments. Firstly, in response to the suggestions made by the experts, we present the initial selection of multivariate analysis of variance (ANOVA) to test the effect of a single independent variable on the dependent variable. Although some data do not follow a normal distribution, ANOVA can mitigate the impact of non-normality. Thus, the results serve as supplementary explanations. Secondly, the concerns raised by the experts regarding the normality of the data may stem from unmet ANOVA prerequisites, which can result in incomplete calculations. Finally, after careful consideration, we have decided to exclude the ANOVA content in order to avoid potential errors, retaining only the results from the independent samples t-test, which also supported by previous literature [1,2].
- Archiza, B.; Andaku, D.K.; Caruso, F.C.R.; Bonjorno, J.C.; Oliveira, C.R.d.; Ricci, P.A.; Amaral, A.C.d.; Mattiello, S.M.; Libardi, C.A.; Phillips, S.A.; et al. Effects of inspiratory muscle training in professional women football players: a randomized sham-controlled trial. Journal of Sports Sciences 2017, 36, 771-780, doi:10.1080/02640414.2017.1340659.
- Martín-Sánchez, C.; Barbero-Iglesias, F.J.; Amor-Esteban, V.; Martín-Sánchez, M.; Martín-Nogueras, A.M. Long-Term Effects of Inspiratory Muscle Training in Institutionalized Elderly Women: A Double-Blind Randomized Controlled Trial. Gerontology 2023, 69, 30-36, doi:10.1159/000522010.
After “however, several limitations need to be addressed”. Please list all the limitations of your study.
Response: Thank you for your valuable comments. We have enhanced the content of this section to ensure its completeness. Some explanations were added to related paragraph. Please check the updated Line 404-409.
Thank you for your review of the article and your valuable comments. Your feedback has not only helped me improve the content of the paper but has also deepened my understanding of the research field.

Reviewer 2 Report
Comments and Suggestions for Authors
Article edited and its can published
Author Response
Thank you for your review of the article and your valuable comments. Your feedback has not only helped me improve the content of the paper but has also deepened my understanding of the research field.